# Using Signal Features of Functional Near-Infrared Spectroscopy for Acute Physiological Score Estimation in ECMO Patients

**DOI:** 10.3390/bioengineering11010026

**Published:** 2023-12-26

**Authors:** Hsiao-Huang Chang, Kai-Hsiang Hou, Ting-Wei Chiang, Yi-Min Wang, Chia-Wei Sun

**Affiliations:** 1Division of Cardiovascular Surgery, Department of Surgery, Taipei Veterans General Hospital, Taipei 11217, Taiwan; 2Department of Surgery, School of Medicine, College of Medicine, Taipei Medical University, Taipei 11031, Taiwan; 3Biomedical Optical Imaging Lab, Department of Photonics, Institute of Electro-Optical Engineering, College of Electrical and Computer Engineering, National Yang Ming Chiao Tung University, Hsinchu 30010, Taiwanchiaweisun@nycu.edu.tw (C.-W.S.); 4Institute of Biomedical Engineering, College of Electrical and Computer Engineering, National Yang Ming Chiao Tung University, Hsinchu 30010, Taiwan; 5Medical Device Innovation and Translation Center, National Yang Ming Chiao Tung University, Taipei 112304, Taiwan

**Keywords:** near-infrared spectroscopy (NIRS), extracorporeal membrane oxygenation (ECMO), microcirculation, acute physiologic and chronic health evaluation II (APACHE II) scoring system, support vector machine (SVM)

## Abstract

Extracorporeal membrane oxygenation (ECMO) is a vital emergency procedure providing respiratory and circulatory support to critically ill patients, especially those with compromised cardiopulmonary function. Its use has grown due to technological advances and clinical demand. Prolonged ECMO usage can lead to complications, necessitating the timely assessment of peripheral microcirculation for an accurate physiological evaluation. This study utilizes non-invasive near-infrared spectroscopy (NIRS) to monitor knee-level microcirculation in ECMO patients. After processing oxygenation data, machine learning distinguishes high and low disease severity in the veno-venous (VV-ECMO) and veno-arterial (VA-ECMO) groups, with two clinical parameters enhancing the model performance. Both ECMO modes show promise in the clinical severity diagnosis. The research further explores statistical correlations between the oxygenation data and disease severity in diverse physiological conditions, revealing moderate correlations with the acute physiologic and chronic health evaluation (APACHE II) scores in the VV-ECMO and VA-ECMO groups. NIRS holds the potential for assessing patient condition improvements.

## 1. Introduction

The human circulatory system serves as the source of energy and nutrients for our bodies. This system facilitates the transfer of oxygen, carbon dioxide, and nutrients between cells, while eliminating metabolic waste. The circulatory system primarily comprises the lungs, heart, arteries, veins, and microcirculation. Microcirculation refers to the network of the tiniest blood vessels, with blood vessels having a diameter smaller than 100 μm [1]. This microvasculature is responsible for substance exchange between the blood and cells, including oxygen, carbon dioxide, and nutrients. The functioning of microcirculation is mainly regulated by endothelial cells, which control aspects such as blood vessel dilation and constriction, vascular permeability, local blood flow control, and anticoagulation. When the functions of endothelial cells are disrupted, it can lead to cardiovascular diseases.

Research using optical technology has shown the capability of observing vascular images on the skin’s surface, and categorized abnormalities in microcirculation into five groups [2].

Class I (all capillaries stagnant): This phenomenon was observed in certain septic shock patients following the excessive use of vasopressors to normalize blood pressure, potentially hindering microcirculatory flow. The absence of capillary flow in this class would result in tissue hypoxia.

Class II (capillaries with flowing red blood cells next to capillaries with no flow): Predominantly observed during cardiopulmonary bypass surgery, this involves disruptions in normal blood flow in micro-veins. This leads to variations in blood flow between two micro-vessels, where one may experience a lack of blood flow, resulting in localized tissue hypoxia [3]. Despite the presence of higher blood hemoglobin saturation, the second vessel exhibits a reduction in the gas exchange surface area [4], a characteristic often observed in patients utilizing an extracorporeal membrane oxygenation (ECMO) device. These circumstances pose a potential threat to tissue perfusion in affected patients.

Class III (stagnant capillaries next to flowing capillaries): This is noted in conditions such as sepsis, reperfusion injury, sickle cell crisis, and malaria. Regions with stagnant capillaries are likely to be hypoxic due to insufficient circulation.

Class IV (hyperdynamic and stagnant capillaries): This is evident in resuscitated hyperdynamic septic patients. The presence of stagnant capillaries contributes to regional tissue hypoxia.

Class V (hyperdynamic flow in all vessels): This is identified in resuscitated sepsis and extreme exercise. It remains unclear whether this leads to hypoxia or signifies shunting from other organs. 

In summary, Classes I, II, III, and IV are characterized by areas of stagnant capillary flow, rendering them incapable of delivering oxygen and resulting in regional tissue hypoxia. The impact of Class V is less distinct based on the provided information. It is noteworthy, however, that our research focuses specifically on the investigation of Class II in this paper. Monitoring these microcirculatory changes could prove valuable in guiding more targeted shock resuscitation.

ECMO is a life-support system used in clinical settings. ECMO plays a pivotal role in saving lives during critical situations and typically involves establishing peripheral vascular access, which can be achieved through various routes such as the femoral artery, femoral vein, or internal jugular vein. ECMO primarily operates in two modes: veno-venous (VV-ECMO) and veno-arterial (VA-ECMO). In VV-ECMO mode, healthcare professionals withdraw venous blood, oxygenate it, and then return it to the patient’s venous system. This mode is appropriate for patients with normal heart function but impaired lung function, such as adults with acute respiratory distress syndrome (ARDS). VV-ECMO effectively provides respiratory support while minimizing the complications associated with arterial cannulation [5,6]. Conversely, in the VA-ECMO mode, venous blood is withdrawn from the patient and oxygenated through an oxygenator before being returned to the patient’s arterial system. This device provides life support for patients with heart and lung failure, sepsis, or cardiogenic shock using ECMO technology.

During the process of using the ECMO treatment, clinical physicians adjust parameters such as the temperature, blood pump speed, and oxygen concentration based on various physiological indicators such as body temperature, blood pressure, heart rate, blood oxygen saturation, coagulation time, mean arterial pressure, and cardiac output to assist patients in achieving optimal circulatory conditions. In recent years, ECMO-assisted therapy has significantly improved patient survival rates. However, as the duration of ECMO use increases, the risk of complications also rises. These complications may include hemolysis due to rapid adjustments in the blood pump speed and the formation of blood clots in the ECMO circuit due to the increased contact between blood and the gas exchange membrane, leading to the activation of coagulation factors and inflammatory mediators. This can increase the risk of peripheral tissue ischemia and the need for amputation. Therefore, patients need continuous administration of anticoagulants during ECMO use to prevent clot formation [6,7]. The administration of anticoagulants during ECMO placement may reduce the platelet count and increase the bleeding risk, particularly in surgical sites, the digestive tract, and intracranial regions [8]. Despite the limitations and possible complications of using physiological indicators as monitoring standards in clinical practice, these parameters still have limitations, and complications cannot be entirely avoided. For instance, neurological problems, stroke, renal replacement therapy, and infections occur at relatively high rates [9,10]. The occurrence of complications increases the mortality rate. Therefore, an increasing number of research teams are dedicated to studying how to prevent complications and exploring various factors contributing to complications and mortality [11,12,13] to enhance the survival rate of ECMO patients. In the past, many research teams have developed scoring systems like SAVE, PRESET, and ENCOURAGE, which utilize logistic regression to predict the survival and mortality rates of ECMO patients based on various clinical physiological parameters, enabling clinical physicians to have better access to assess the patient’s condition [14,15,16]. Currently, when patients are undergoing ECMO, clinical physicians can only adjust the ECMO mode based on fundamental physiological signs and the relevant dosage of medications. Nevertheless, using this method to detect changes in the hemodynamics of peripheral tissues in patients is relatively sensitive.

Near-infrared spectroscopy (NIRS) is a non-radiating, non-invasive, and real-time monitoring technique. The objective of this study is to utilize NIRS to monitor blood perfusion in the lower limb of ECMO patients. The goal is to establish the correlation between the optical observation of blood oxygen dynamics during ECMO pump speed adjustments and the corresponding physiological indicators. Researchers have recently applied this technology to identify postoperative complications arising from insufficient local blood flow. In 2012, Wong et al. first used NIRS devices to measure cerebral and limb blood perfusion in ECMO patients [17]. They integrated this technology into treatment protocols, indicating that immediate interventions are necessary when the regional oxygen saturation (rSO_2_) drops below 40% of the baseline or rises above 25% [18,19]. Lamb et al. confirmed that continuous monitoring using NIRS can detect ischemia in the limbs of ECMO patients [20]. In addition to the interest in complication detection, research on the application of NIRS for monitoring systemic circulation has been growing. In 2020, our laboratory conducted a study utilizing NIRS technology to examine the adjustment of the blood pump speed in ECMO patients as part of clinical physiological parameter monitoring. During the research, we also observed perfusion in peripheral tissues. Our preliminary findings have been published in the *Journal of Biophotonics* [21], highlighting the potential of NIRS in providing real-time insights into patients’ physiological responses during ECMO treatment.

At Taipei Veterans General Hospital, where the experiments were conducted, there are an average of 200–250 ECMO patients annually. Drawing from our long-term experience in caring for a large number of ECMO patients, we have observed that the natural physiological response of the human body is to prioritize blood circulation to central organs in cases of severe injury or shock (with the exception of the kidneys, which need to retain sufficient fluid in the body). Peripheral tissue circulation is often reduced or sacrificed first, and the reduction in blood perfusion is most evident in the distal regions, particularly the feet. In ECMO patients, good blood circulation in the distal legs suggests a higher chance of survival, whereas poor circulation is associated with a higher likelihood of mortality. Due to challenges in measuring SPO2 or NIRS changes in the foot and toe regions, such as insufficient muscle mass, low temperature, and intense microvascular constriction, the calf was chosen as the measurement site for NIRS in our serial studies.

Beyond the technical applications, we believe that this study holds significant implications for enhancing medical procedures in clinical practice. Through the implementation of non-invasive optical technology, we can observe real-time changes in patients’ blood oxygen levels, gaining further insights into the severity of the disease.

For patients in the high-severity group, our study provides a valuable indicator for physicians when considering adjustments in ECMO support or medication dosages. Conversely, for patients with a lower disease severity, physicians may contemplate gradually reducing the ECMO rotation speed, allowing patients to rely more on their circulatory system, potentially leading to the complete cessation of ECMO treatment.

In essence, this research aims to equip clinical physicians with precise predictions of critical conditions in severe patients, thereby reducing the risk of complications and minimizing the use of healthcare resources. The technical advancements in NIRS not only contribute to our understanding of ECMO dynamics but also offer tangible benefits in improving patient outcomes and optimizing clinical decision making.

## 2. Materials and Methods

This study was approved by the Institutional Review Board (IRB) of Taipei Veterans General Hospital (TVGHIRB-2019-02-007AC). The subjects were adults (age ≥ 20 years) from Taipei Veterans General Hospital who were evaluated by cardiothoracic surgeons and deemed to need ECMO assistance therapy. Exclusion criteria included patients receiving central ECMO, patients weighing less than 45 kg, and patients not suitable for near-infrared blood oxygen monitoring.

### 2.1. Patients

This research involved patients who underwent treatment with VA-ECMO and VV-ECMO. Patients who met the following conditions were not included in the study: (1) the speed of the membrane oxygenator was too high or too low, resulting in circuit shaking; or (2) the blood oxygen saturation (SpO_2_) was insufficient. A total of 40 individuals were included in the study, divided into two categories based on the method of membrane oxygenator insertion: 22 VV-ECMO patients and 18 VA-ECMO patients (Table 1).

### 2.2. NIRS Measurement and Analysis

Our experiment uses the PortaLite system (Artinis, Netherlands) as our measurement instrument. The measuring device and processing flowchart are shown in Figure 1. We place two PortaLite systems on the calf of the subjects, beginning at approximately 60% of the calf’s length from above the ankle, to measure the hemodynamic response (Figure 1a). This instrument consists of three dual-wavelength LED light sources and one sensor. The wavelengths of the LED light sources are 760 nm and 850 nm in the near-infrared range, and the distances between the light sources and the sensor are 30 mm, 35 mm, and 40 mm, respectively (Figure 1b). The probe’s maximum sampling rate is 50 Hz, and we transmitted the acquired light-intensity data to the computer via Bluetooth and processed it at a sampling rate of 25 Hz. Using the dedicated software Oxysoft (version 3.0) for PortaLite and the modified Beer–Lambert law (MBLL) [22], we calculated the relative concentration changes of oxygenated hemoglobin (HbO_2_) and deoxygenated hemoglobin (HHb) from the obtained blood oxygen signals. After filtering and normalization, we input these signals into our machine-learning algorithm.

#### 2.2.1. Measurement Protocol

At the start of the measurement phase, we adjust the blood pump speed of the ECMO system and conduct a 15 min assessment to ensure stable blood oxygen signals for the subject. Depending on whether it is VV-ECMO or VA-ECMO, we make speed adjustments in increments of 300 or 500 revolutions per minute (rpm), respectively. After the initial 15 min of data collection, we decrease the speed by one unit for a 10 min measurement. Then, we increase the speed by one unit for a 10 min measurement, repeating this process in the following two stages, i.e., each lasting 10 min, with one-unit increases in speed. Finally, we meticulously reset the speed to its initial value, enabling us to conduct a comprehensive 15 min measurement session. The total experiment duration is approximately 70 min (Figure 1c).

#### 2.2.2. Microcirculation Monitoring Results

For all VV-ECMO patients, the access points are the right internal jugular vein and femoral vein. For all VA-ECMO patients, the access points are the femoral vein and femoral artery. Different access points may impact NIRS measurements. However, these two patient groups will be compared separately.

Simultaneously, in terms of microcirculation monitoring results, whether it is VA- or VV-type ECMO cannulation, the typical setup involves one end of the tubing connected to the patient’s leg vein or femoral artery. Consequently, the patient’s lower limb can be divided into the cannulation and the non-cannulation sides. The tubing is placed within the blood vessels on the cannulation side, so ECMO setup significantly impacts the blood perfusion in the peripheral tissues. It does not allow effective monitoring of the patient’s overall circulatory function. Hence, this study primarily focuses on the changes in blood flow on the non-cannulation side of the subjects (Figure 1d) [21].

#### 2.2.3. Statistical Analysis

In NIRS measurements, raw data often contain physiological interferences, including periodic respiration (0.15–0.4 Hz) and heartbeat (0.4–1.6 Hz) [23,24]. Consequently, filtering the raw signals is essential to extract meaningful blood oxygen information (Figure 2). This study employed a low-pass filtering approach to isolate the blood oxygen signal. We utilized signal.filtfilt from the Python Scipy module, ensuring that the filtered signal preserves the essential characteristics of the original signal without introducing phase delays. Specifically, we employed a fourth-order Butterworth low-pass filter with a cutoff frequency set at 0.1 Hz [25]. These filter parameters were selected to effectively mitigate physiological noise sources such as respiration and heartbeat, enhancing the accuracy of blood oxygen data analysis (Figure 2b). Subsequently, we normalized the signal after low-pass filtering using MinMaxScaler from the Python Scikit module. Normalization aims to eliminate data disparities caused by individual differences among subjects. We rescaled the filtered values to a range between 0 and 1 while preserving their original proportions for subsequent evaluation and comparison (Figure 2c).

#### 2.2.4. Feature Extraction

After preprocessing the signals mentioned above, we extract features (Feature 3) from the signals of oxygenated blood (HbO_2_), deoxygenated blood (HHb), total hemoglobin (HbT), and tissue saturation index (TSI) for each stage mean values. In addition, HbO_2_, HHb, and HbT are utilized for the other three features. The detailed description is as follows:Stage mean: The arithmetic average of oxygenation information for different stages (Figure 3a), a total of 24 features (6 stages × 4 signals);Stage activation: The difference between the mean value of each stage and the mean value of the baseline stage (Figure 3b), a total of 15 features (5 stages × 3 signals);Stage mean difference: Calculation of the difference between the mean of each stage and the mean of the previous stage (Figure 3c), a total of 15 features (5 stages × 3 signals);Stage slope: The slope of the oxygenation information from the beginning to the end of different task stages (Figure 3d), a total of 18 features (6 stages × 3 signals).

Based on the features introduced above, a total of 72 features are obtained, while there were 3 duplicate features: Hence, we obtained 69 filtered and normalized features. These extracted features will serve as inputs for training and prediction in our machine-learning model.

Support vector machine (SVM) is a supervised learning method suitable for handling scenarios with small sample sizes, non-linear data, or high-dimensional data. Its underlying principle revolves around finding an effective decision boundary that separates samples into distinct classes [26,27,28]. Consequently, SVM performs exceptionally well in both classification and regression tasks. We utilize kernel functions that transform data points into higher-dimensional spaces for classification to address non-linearity and high-dimensional data. In addition, we specifically employ the radial basis function (RBF) kernel. 

Cross-validation is an indicator for evaluating the generalization ability of machine-learning models. This process entails assessing the model’s performance on unfamiliar data to address concerns like overfitting and data selection bias. A prevalent approach to cross-validation is k-fold cross-validation, with our approach utilizing 5-fold cross-validation.

Taking 5-fold cross-validation as an example (Figure 4), our dataset is randomly divided into 5 groups. One group is designated as the validation set, while the remaining 4 groups form the training set. In each iteration, a cross-validation (CV) value is obtained, representing accuracy. This iterative process continues until each group has been utilized as the validation set. Subsequently, the average accuracy across the 5 validation sets is compared with the actual model test results, serving as an evaluative metric for the model’s performance.

We utilized confusion matrices and 5-fold to perform model performance and evaluate model generalization.

#### 2.2.5. Disease Severity Assessment Scale

We utilized the acute physiologic and chronic health evaluation score (APACHE II) scale to assess the physiological condition of patients. The APACHE scoring system is commonly employed to evaluate the mortality risk of intensive care unit (ICU) patients. This system comprises two distinct components: the acute physiologic score and the chronic health evaluation. These components are determined based on 12 physiological parameters, age, and the presence of chronic medical conditions. The scoring system ranges from 0 to 71, with higher scores indicating a more severe illness.

In 1985, Knaus et al. established a system that combines APACHE II scores, disease-specific weighting, and surgical status to predict patient mortality (Table 2) [29]. In this study, a mortality rate of approximately 50% served as the threshold. Patients with an APACHE II score of 24 or lower were labeled as the low-score group (L), while those with scores exceeding 24 were labeled as the high-score group (H). 

## 3. Results

### 3.1. Classification Results of High and Low APACHE II Score Groups

This study labeled APACHE II scores equal to or less than 24 as the low-score group (L), while scores greater than 24 were the high-score group (H). In the VV-ECMO group, there were 44 data points (L14/H30), and, while in the VA-ECMO group, there were 41 data points (L22/H19). All NIRS measurements were conducted simultaneously during a consistent time period (10 am to 11 am) to ensure uniformity across assessments. This approach aims to enhance the precision of recording and analyzing changes in disease severity or APACHE scores. 

In machine-learning classification, we integrated both NIRS and non-NIRS features. NIRS signals represent the blood oxygen characteristics of patients during task phases, while non-NIRS signals represent clinical data such as age and body mass index (BMI). There were 69 features in the machine-learning model for training and prediction. However, too many or irrelevant feature parameters during model training can lead to overfitting, compromising the model’s generalization ability. To address this issue, this study employed independent-sample *t*-tests to statistically assess the differences in mean values of these feature parameters between the high-score and low-score groups, aiming to identify features that effectively contribute to classification.

The results of our statistical testing for NIRS and non-NIRS feature parameters in both the VV-ECMO and VA-ECMO groups are presented in Table 3 and Table 4. These tables are arranged in ascending order based on *p*-values. In our analysis, when the independent-sample *t*-test yielded a *p*-value less than 0.05, we considered it a statistically significant difference and marked it with a single asterisk (*). A *p*-value less than 0.01 indicates a highly significant difference, denoted with double asterisks (**).

Figure 5 displays two sets of significant feature parameters selected from the VV-ECMO and VA-ECMO groups and shows their distributions in the low and high subgroups. For the VV-ECMO population classification, we constructed an SVM model using NIRS features selected through statistical tests that exhibited significant differences. The best-performing model included two feature values: the average difference in oxygenated hemoglobin concentration during Task Stage 3 and the average value of total hemoglobin concentration during Task Stage 2. Figure 5a,b depict the data distribution of these two features in the high-scoring and low-scoring subpopulations within the VV-ECMO group, while Figure 5c,d display the data distribution of age and BMI in the high-scoring and low-scoring subpopulations within the same group. In the classification results of the VA-ECMO population, the better-performing model relies on two feature parameters: the slope of oxygenated hemoglobin concentration during Task Stage 2 and the average value of deoxygenated hemoglobin concentration during Task Stage 5. Figure 5e,f showcase the data distribution of these two classification features in the high-scoring and low-scoring subpopulations within the VA-ECMO group. In contrast, Figure 5g,h depict the data distribution of age and BMI in the high-scoring and low-scoring subpopulations within the same group. 

### 3.2. SVM Classification Results

The data allocation for SVM is presented in Table 5. In this study, there are 44 data points for the VV-ECMO group and 41 data points for the VA-ECMO group. To determine the data allocation for training and testing, we considered the proportions of each group’s data in the original dataset. Specifically, 70% of the data from both groups were designated for training, while the remaining 30% were allocated for testing. To ensure the integrity of the training process and prevent any influence from the testing data that could compromise the model’s generalization ability, we conducted statistical tests on the initial feature parameters using only the training data. The testing data were reserved for evaluating the model’s performance.

### 3.3. NIRS Signals

#### 3.3.1. VV-ECMO Classification Model

Figure 6 depicts the classification model for the VV-ECMO population based on two-dimensional features. Figure 6a illustrates the model’s performance on the training data, achieving an 83.9% accuracy. In Figure 6b, we can observe the results when applying the trained model to the testing data, resulting in a 76.9% accuracy. In these figures, red dots represent data points from the low-score subgroup, while blue dots represent data points from the high-score group.

While these results show that the decision boundary can roughly distinguish between the two VV-ECMO groups, there is still a mixture of high- and low-score data points within the training model, leading to classification errors. Upon closer examination of these erroneous data points, it becomes evident that about half of them have APACHE II scores near the boundary between the high and low groups, contributing to the reduction in testing accuracy to 76.9%.

To assess the classification performance and generalization ability of the two ECMO classification models, we used a normalized confusion matrix, depicted in Figure 6c, which yielded a sensitivity of 78% and a specificity of 75%. In the results of the five-fold cross-validation (Table 6), the average accuracy of the cross-validation is 77.2%, with a standard deviation of 7.9%, indicating that the model exhibits high stability.

#### 3.3.2. VA-ECMO Classification Model

The classification model for the VA-ECMO population based on two-dimensional features is depicted in Figure 7a, while Figure 7b illustrates the results of testing data when input into the trained model. The training accuracy is 85.7%, and the testing accuracy is 84.6%. 

The figures reveal that this model’s decision boundary effectively separates the VA-ECMO population data points. In contrast to the VV-ECMO model, the data distribution of the two groups is relatively straightforward, resulting in similar training and testing accuracies.

Figure 7c presents the normalized confusion matrix, indicating a sensitivity of 83% and a specificity of 86%. In the results of the five-fold cross-validation (Table 7), the average accuracy of the cross-validation is 83.1%, with a standard deviation of 13.9%, indicating that the model has strong generalization capabilities and can make excellent distinctions for unknown data.

### 3.4. Incorporating NIRS Signals with Clinical Parameters

To enhance the classification performance by leveraging additional high-dimensional information, we integrated the classification models for the VV-ECMO and VA-ECMO populations previously mentioned with non-NIRS features (BMI and age). To achieve this, we employed principal component analysis (PCA) to reduce the dimensionality of the four-dimensional features to two dimensions and, subsequently, incorporated them into the SVM model (Figure 8 and Figure 9).

#### 3.4.1. VV-ECMO Classification Model

The classification model for the VV-ECMO population attained a training accuracy of 80.6% and a testing accuracy of 84.6%, as illustrated in Figure 8a,b. These accuracies closely mirror those of the model that did not incorporate clinical parameters previously.

Consistent with the earlier model, the misclassified data points demonstrate a recurring pattern, with roughly half of the data exhibiting APACHE II scores near the boundary that distinguishes the two groups. Given the present distribution of the dataset, where there are more data points in the low-score group compared to the high-score group, achieving an accuracy of 68.2% by categorizing all data as the high-score group suggests that the current model possesses predictive capability.

The confusion matrix reveals a reduced proportion of classification errors compared to the previous model (Figure 8c), with a sensitivity of 89% and a specificity of 75%. In the results of the five-fold cross-validation (Table 8), the average accuracy of the cross-validation is 77.2%, with a standard deviation of 13.7%.

#### 3.4.2. VA-ECMO Classification Model

In the VA-ECMO population, with the inclusion of clinical parameters, the classification results in Figure 9 show a more distinct separation between the two categories. The training accuracy and testing accuracy of the classification model are 92.2% and 84.6%, respectively, as shown in Figure 9a,b. The results of the normalized confusion matrix, displayed in Figure 9c, indicate improved model generalization compared to the previous model, with a sensitivity of 83% and a specificity of 86%. In the results of the five-fold cross-validation (Table 9), the average accuracy of the cross-validation is 85%, with a standard deviation of 16.3%.

## 4. Discussions

This study aims to differentiate between low- and high-scoring groups in the VV-ECMO and VA-ECMO populations by combining near-infrared spectroscopy (NIRS) with the APACHE II severity-of-disease scoring system. In the classification results for the VV-ECMO population, we constructed an SVM model using NIRS features that exhibited significant differences based on statistical tests. The top-performing classification model consisted of two feature values, each associated with stages where rotational speed adjustments were minimal. Figure 5a,b illustrate the data distribution of these two features in the high-scoring and low-scoring populations within the VV-ECMO group. On average, in the high-scoring group, there was a significant increase in the average change in oxygenated hemoglobin concentration during Task Stage 3 when the pump speed was increased by one unit compared to the average change in oxygenated hemoglobin concentration during Task Stage 2. This observation suggests that the high-severity group requires higher pump speeds to achieve better oxygenation at Stage 3.

Additionally, in the high-scoring group, the average increase in total hemoglobin concentration during Task Stage 2 exceeded that in the low-scoring group. This finding implies that the high-severity group has relatively poorer lung circulation. When the pump speed was increased by one unit from the minimum level at Stage 1, peripheral blood vessels exhibited a more pronounced expansion.

Incorporating clinical parameters into the VV-ECMO population can enhance the model’s sensitivity, especially for accurately diagnosing highly severe patients. Figure 5c,d depict the distribution of age and BMI data in the high-score and low-score groups within the VV-ECMO cohort. There were no significant differences in age between these two groups. While the APACHE II scale does not explicitly include BMI as an indicator, BMI, nevertheless, reflects APACHE II scores. We observed that patients in the low-score group had higher overall BMI. Recent studies have indicated that a patient’s BMI can influence ICU survival rates. Research findings have suggested that underweight patients have a higher risk of in-hospital mortality, whereas overweight patients have a lower mortality risk [30]. Patients with a slightly higher BMI tend to store more calories, making them more resilient when faced with severe illnesses and providing them with more energy to combat diseases, resulting in higher survival rates. In the future, combining BMI indicators with the NIRS system may aid in assessing the ultimate outcomes of ECMO patients, whether they survive or succumb.

Regarding the classification outcomes in the VA-ECMO population, the better-performing classification models correlate with stages where the patient’s internal microcirculation is significantly affected by adjustments in the blood pump speed. Figure 5e,f illustrates the data distribution of two classification features within the VA-ECMO cohort among the high-score and low-score groups. In the low-score group, there is a more pronounced trend in the change of oxygenated blood concentration during Task Stage 2 compared to the high-score group. This finding is because patients in the low-score group have relatively better circulatory function, resulting in a higher increase in oxygenated blood concentration when increasing the blood pump speed during Task Stage 2. Conversely, in the high-score group, when reducing the pump speed by two units during Task Stage 5, a substantial reduction in ECMO support occurs. During this time, peripheral tissues experience extreme hypoxia, leading to a higher change in hypoxic blood concentration during Task Stage 5 compared to the low-score group. We incorporate clinical parameters into the VA-ECMO population with the expectation of achieving improved performance in training the model and enhancing its generalization capability. Figure 5g,h depicts the distribution of age and BMI data among high-score and low-score groups within the VA-ECMO cohort. In contrast to the VV-ECMO population, BMI did not show significant differences between the high-score and low-score groups. However, in the low-score group, patients tended to be younger. This outcome finds its explanation in the classification labels’ foundation on APACHE II scores, incorporating patient age as one of the scoring criteria. Therefore, older patients may receive higher APACHE II scores, which is also why patients in the low-score group generally have a younger age.

From Figure 6 to Figure 9, we can observe certain phenomena. In terms of model accuracy, whether using NIRS oxygenation features for classification or optimizing the model by incorporating clinical parameters, the classification model for the VA-ECMO population outperforms that of the VV-ECMO population. Overall, the VA-ECMO population’s model demonstrates superior generalization. The two data categories exhibit a clear separation in the two-dimensional feature distribution plot, indicating that they do not overlap. Consequently, the decision boundary effectively distinguishes the majority of data points between these two categories. Both sensitivity and specificity highlight the VA-ECMO population’s model as having superior discriminative abilities compared to the VV-ECMO population’s model.

The VV-ECMO population’s model exhibits relatively poorer performance due to several factors. Firstly, there was an imbalance in the number of samples between the two VV-ECMO subgroups, leading to an overemphasis on the majority category during model training, resulting in a reduced performance for the minority category samples. Secondly, based on the statistics of APACHE II scores (Figure 10a,b), a significant number of data points near the APACHE II score of 24 were observed within the VV-ECMO population. Within this range, the severity of illness is quite similar, and, as a result, any misclassification of data points in this interval significantly impacts the overall classification accuracy. In contrast, VA-ECMO is less affected by such scenarios.

In future research directions, we anticipate that, with an increase in patient sample size, traditional analytical methods such as multivariate models incorporating classical biomarkers can be employed. These can be compared with established prognostic scores like APACHE II and SOFA to provide alternative predictive techniques beyond traditional analysis. Due to the preliminary nature of this series of studies and the uncertainty surrounding the future value of big data, coupled with the limited number of patients, this article does not present statistical analyses for the small patient subset.

This near-infrared spectroscopy (NIRS) study demonstrates significant differences in patient prognostic scoring compared to traditional methods. In addition to offering alternative models for predicting patient outcomes, it also aids ICU physicians in timely adjustments of ECMO settings, such as early adjustments of ECMO support when patients show improvement, thereby enhancing the quality of ECMO care.

Despite the early literature on ECMO mentioning risk factors for predicting patient outcomes, the predictive capability for individual patients remains insufficient. Based on the analysis results presented in this paper, we hypothesize that NIRS may have the potential to surpass previous multivariate risk factor analysis models in terms of the predictive capability for individual patients.

## 5. Conclusions

In this study, we used NIRS-monitoring devices to collect blood oxygenation data from the lower limbs of patients undergoing ECMO treatment. We extracted meaningful features from a wide range of blood oxygenation data, selected discriminative features through independent-sample *t*-tests and correlation analysis, and incorporated them into machine-learning models to classify populations with high and low disease severity in VV-ECMO and VA-ECMO. In the case of VA-ECMO, we achieved good classification results on both training and testing data, indicating the model’s strong generalization ability. However, for VV-ECMO, the uneven distribution of data resulted in testing data and model validation performance that was not as robust as the training data. Nevertheless, overall, both device modes demonstrated sufficient predictive capability, validating the effectiveness of the optical technology used in diagnosing disease severity in patients.

In the future, we will continue to collect more cases to expand the database and consider incorporating additional assessment scales to enhance the evaluation mechanism of patient conditions. When the number of instances reaches a certain threshold, we can attempt to analyze the relationship between NIRS signals and patient mortality rates, using the APACHE II scale scores and relevant disease diagnoses to predict patient outcomes accurately. Additionally, we can explore implementing deep learning to construct predictive models, replacing manual feature selection. This approach would involve training models with NIRS data from the first few days of ECMO treatment to forecast the future trends in patient blood oxygen signals, providing reliable prognosis references for clinical practitioners and reducing the misuse of healthcare resources.

## Figures and Tables

**Figure 1 bioengineering-11-00026-f001:**
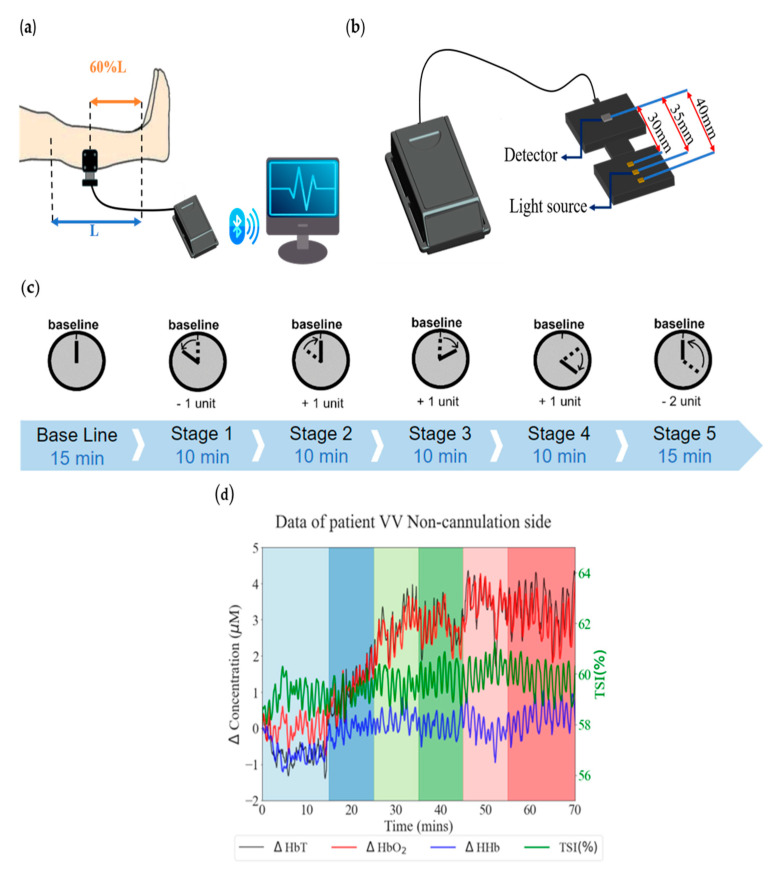
The measuring device and processing flowchart: (**a**) PortaLite instrument appearance, (**b**) instrument placement schematic, (**c**) ECMO measurement experiment flowchart, and (**d**) hemodynamics on the non-cannulation side (VV−ECMO).

**Figure 2 bioengineering-11-00026-f002:**
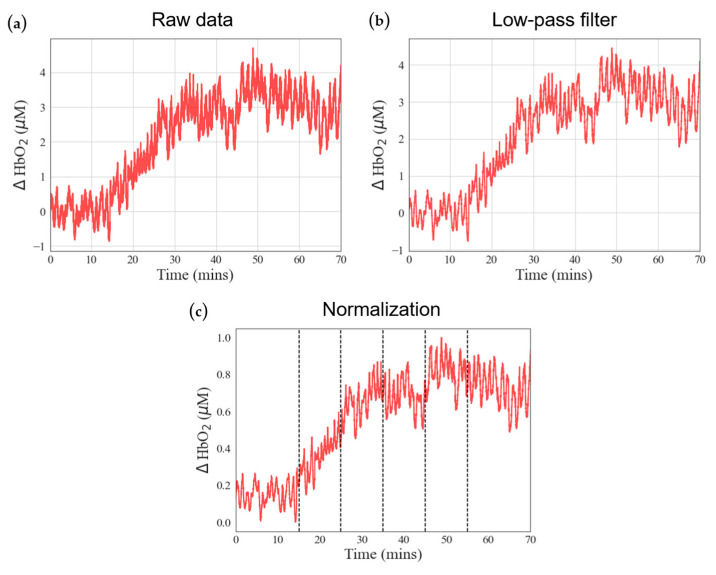
Data preprocessing diagrams: (**a**) raw data signal, (**b**) raw data after adequate low-pass filter, and (**c**) normalization. Red color lines: blood oxygen signals.

**Figure 3 bioengineering-11-00026-f003:**
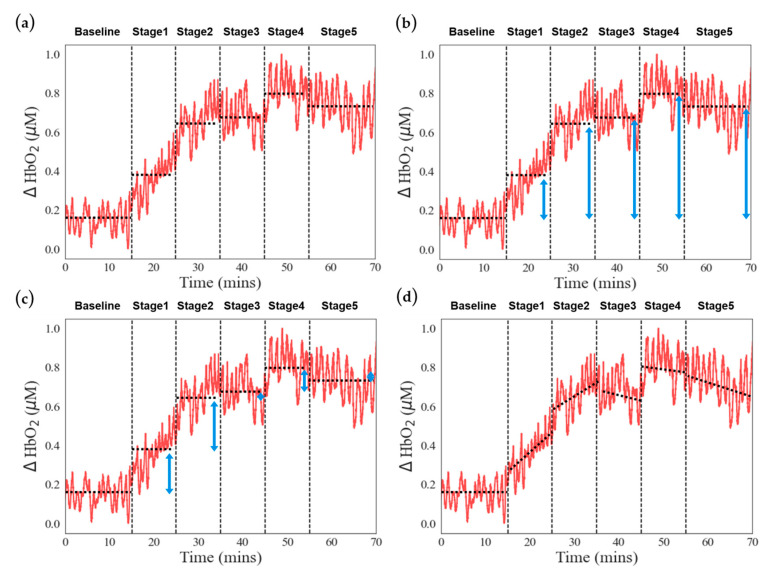
Four feature extractions from signals. Taking the ΔHbO_2_ concentration of one of the subjects during the experiment as an example, including (**a**) stage mean, (black dot line). (**b**) stage activation, (blue double arrow and black dot line). (**c**) stage difference, (blue double arrow and black dot line). (**d**) stage slope, (black dot line). A solid black dotted line distinguishes different stages.

**Figure 4 bioengineering-11-00026-f004:**
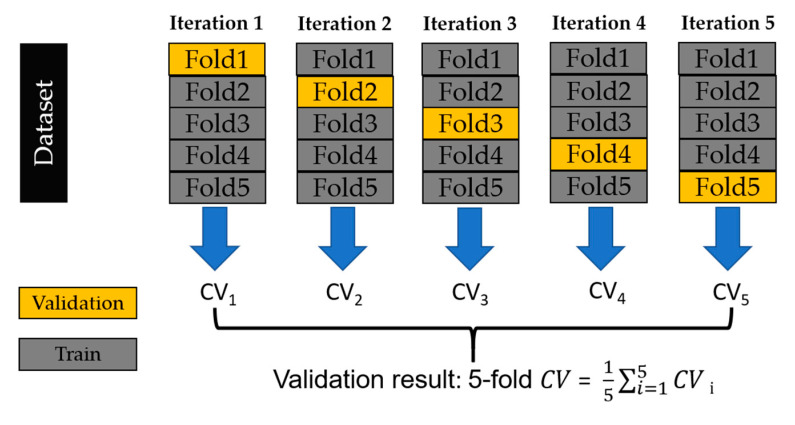
Diagram depicting the 5-fold cross-validation procedure.

**Figure 5 bioengineering-11-00026-f005:**
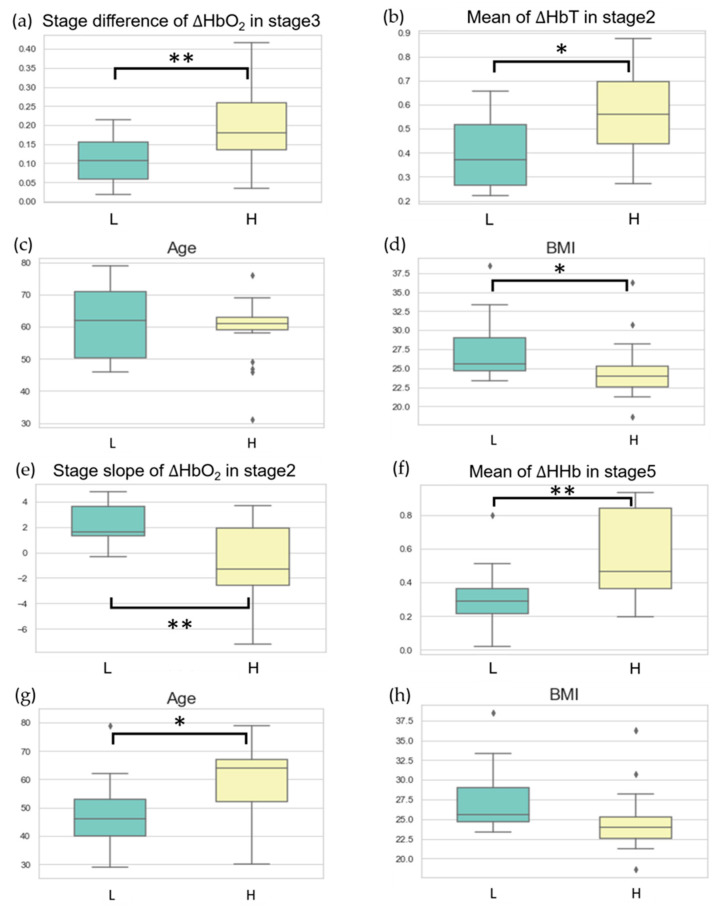
High and low groups’ significant feature parameters selected from the VV−ECMO and VA−ECMO. (**a**,**b**) display box plots of NIRS classification features for VV−ECMO: (**a**) mean difference of HbO_2_ during Task Stage 3, and (**b**) mean value of HbT during Task Stage 2. (**c**,**d**) show box plots of clinical parameters for VV−ECMO: (**c**) age, and (**d**) BMI. (**e**,**f**) present box plots of NIRS classification features for VA−ECMO: (**e**) slope of HbO_2_ during Task Stage 2, and (**f**) mean value of HHb during Task Stage 5. (**g**,**h**) depict box plots of clinical parameters for VA−ECMO: (**g**) age, and (**h**) BMI. *: *p* ≤ 0.05, represents statistical significance, **: *p* ≤ 0.01, represents high statistical significance. Green box plot: distribution of the low-score group (L) across different features. Yellow box plot: distribution of the high-score group (H) across different features.

**Figure 6 bioengineering-11-00026-f006:**
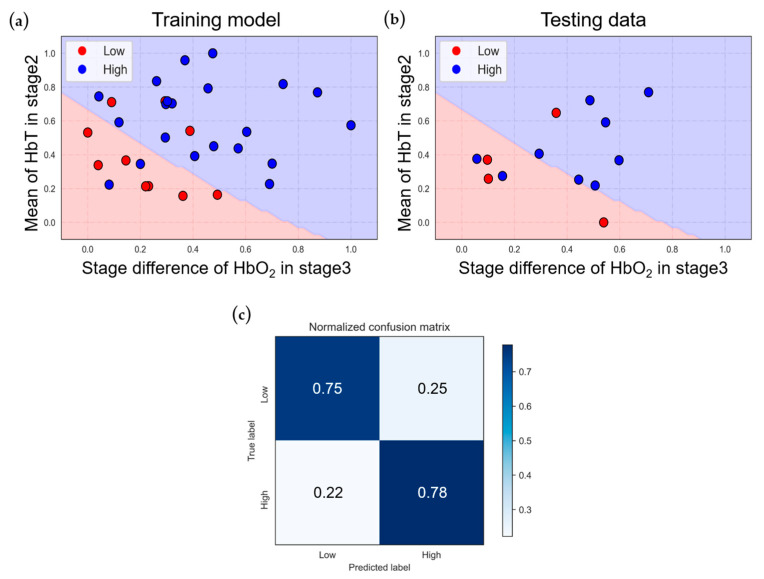
Classification results of VV−ECMO under NIRS features: (**a**) training and (**b**) testing data classification, and (**c**) normalized confusion matrix.

**Figure 7 bioengineering-11-00026-f007:**
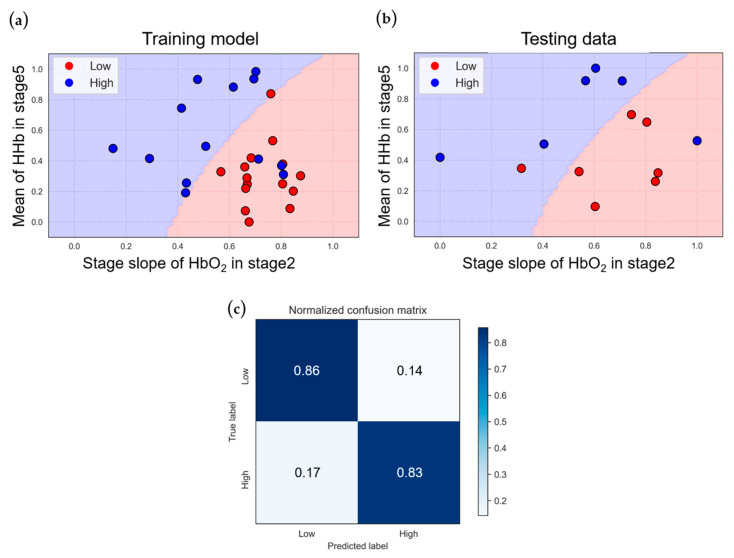
Classification results of VA−ECMO under NIRS features: (**a**) training and (**b**) testing data classification, and (**c**) normalized confusion matrix.

**Figure 8 bioengineering-11-00026-f008:**
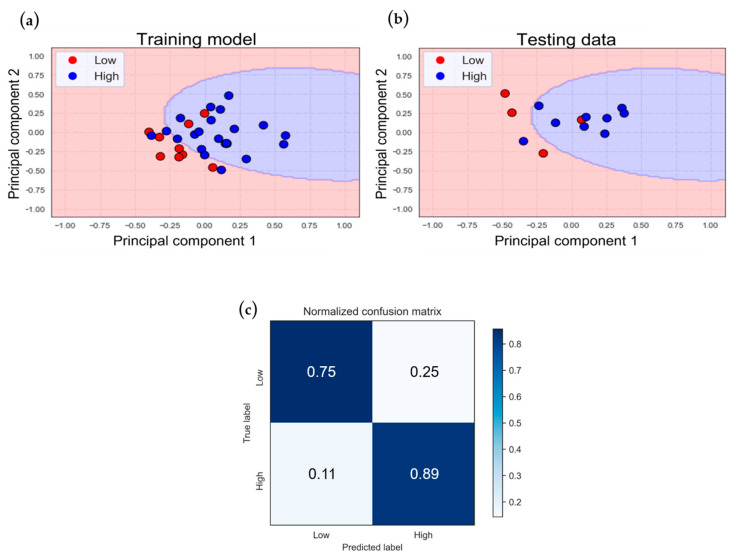
The classification results of VV−ECMO with the addition of clinical parameters in an SVM model: (**a**) training and (**b**) testing data classification, and (**c**) normalized confusion matrix.

**Figure 9 bioengineering-11-00026-f009:**
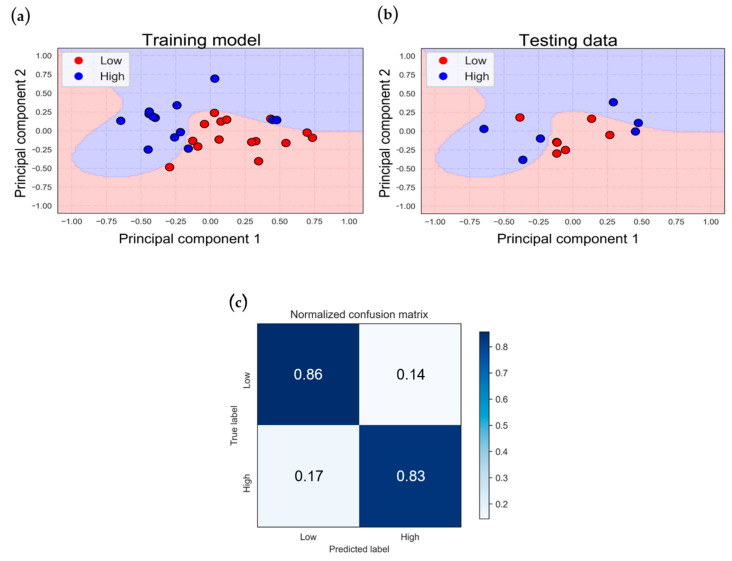
The classification results of VA−ECMO with the addition of clinical parameters in an SVM model: (**a**) training and (**b**) testing data classification, and (**c**) normalized confusion matrix.

**Figure 10 bioengineering-11-00026-f010:**
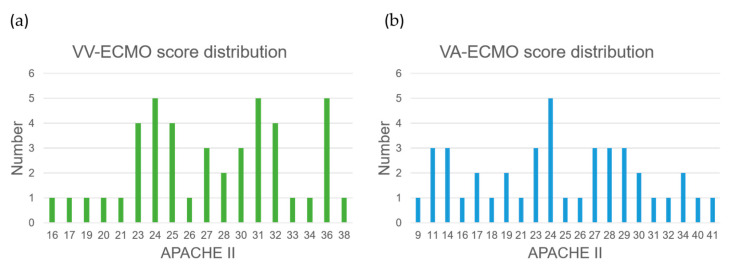
The statistical distribution of APACHE II scores in two groups: (**a**) VV−ECMO group and (**b**) VA−ECMO group. The green bars: the population size of the VV−ECMO group for each corresponding APACHE II score. The blue bars: the population size of the VA−ECMO group for each corresponding APACHE II score.

**Table 1 bioengineering-11-00026-t001:** Information on patients treated with ECMO in this study.

	VV-ECMO (*n* = 22)	VA-ECMO (*n* = 18)
Age (years ± SD)	60 ± 13.8	54.5 ± 16.5
Sex (F/M) ^1^	M12/F10	M13/F15
BMI ^2^ (kg/m^2^ ± SD)	25.9 ± 3.9	26.1 ± 4.8
Set time (days)	14.0 ± 9.2	12.4 ± 7.9
Mortality rate (%)	40.91	77.78

^1^ F, female; M, male; ^2^ BMI, body mass index. BMI is calculated as the weight in kilograms divided by the square of the height in meters.

**Table 2 bioengineering-11-00026-t002:** APACHE II mortality prediction.

APACHE II Score	Mortality
0 to 4	~4%
5 to 9	~8%
10 to 14	~15%
15 to 19	~25%
20 to 24	~40%
25 to 29	~55%
30 to 34	~75%
Over 34	~85%

**Table 3 bioengineering-11-00026-t003:** Independent-sample *t*-tests on NIRS features between VV-ECMO and VA-ECMO groups.

VV-ECMO		VA-ECMO	
Features ^1^	*p*-Value	Features	*p*-Value
**SS_HbO_2__Stage2 ****	0.0079	**SS_HbO_2__Stage2 ****	0.0044
**SD_HbO_2__Stage3 ****	0.0079	**SM_HHb_Stage5 ****	0.0090
**SD_ HbO_2__Stage2 ***	0.0135	**SD_ HbO_2__Stage3 ****	0.0093
**SM_HBT_Stage2 ***	0.0202	**SA_HHb_Stage5 ***	0.0103
**SM_ HbO_2__Stage3 ***	0.0447	**SD_ HbO_2__Stage1 ***	0.0256
**SM_HBT_Stage3**	0.0545	**SA_HHb_Stage4 ***	0.0336
**SA_ HbO_2__Stage3**	0.0545	**SS_HBT_Stage1 ***	0.0341

^1^ SS: stage slope, SD: stage difference, SM: stage mean, SA: stage activation. *: *p* ≤ 0.05, **: *p* ≤ 0.01.

**Table 4 bioengineering-11-00026-t004:** Independent-sample *t*-tests on non-NIRS features between VV-ECMO and VA-ECMO groups.

VV-ECMO		VA-ECMO	
Features ^1^	*p*-Value	Features	*p*-Value
Age	0.5372	Age *	0.0443
BMI *	0.0328	BMI	0.6784

^1^ *: *p* ≤ 0.05.

**Table 5 bioengineering-11-00026-t005:** Data allocation for SVM.

VV-ECMO			VA-ECMO	
	Low	High	Low	High
Total	14	30	22	19
Training data	10	21	15	13
Testing data	4	9	7	6

**Table 6 bioengineering-11-00026-t006:** Five-fold cross-validation results of the classification model for VV-ECMO under NIRS features.

**Fold**	1	2	3	4	5	Average
**Score (%)**	66.7	77.8	77.8	88.9	75.0	77.2 ± 7.9

**Table 7 bioengineering-11-00026-t007:** Five-fold cross-validation results of the classification model for VA-ECMO under NIRS features.

**Fold**	1	2	3	4	5	Average
**Score (%)**	77.8	100	87.5	87.5	62.5	83.1 ± 13.9

**Table 8 bioengineering-11-00026-t008:** Five-fold cross-validation results of the classification model for VV-ECMO with clinical parameters.

**Fold**	1	2	3	4	5	Average
**Score (%)**	77.8	66.7	100	66.7	75.0	77.2 ± 13.7

**Table 9 bioengineering-11-00026-t009:** Five-fold cross-validation results of the classification model for VA-ECMO with clinical parameters.

**Fold**	1	2	3	4	5	Average
**Score (%)**	100	100	75	87.5	62.5	85 ± 16.3

## Data Availability

The data underlying the results presented in this paper are not publicly available in the interest of protecting the human research participants.

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
