# Peer review of "Using Signal Features of Functional Near-Infrared Spectroscopy for Acute Physiological Score Estimation in ECMO Patients"

_bioengineering, 2023, doi:10.3390/bioengineering11010026_

Round 1

Reviewer 1 Report

Comments and Suggestions for Authors

The authors present a study developing an AI model using NIRS and non-NIRS features to predict APACHE scores. Their methods are sound and well described. The overall sample size for their testing is small. 

Overall questions:

Are the NIRS measurements affected if the access point for ECMO is axillary/internal jugular (upper extremity) versus femoral (lower extremity)?

When in the ICU course were the NIRS measurements made? Were the measurements made at the same timepoint for all of the patients? This would impact the severity of APACHE scores. 

Models are presented both for NIRS features and NIRS + non-NIRS features. Given the predictions are better for the combination with non-NIRS (BMI and age), it may be better for the purpose of the manuscript to present the latter results and perhaps put the former in a supplement. In addition, since BMI and age are independent variables that are clinically significant confounders in the APACHE score, it would be optimal to include in the model/algorithm for prediction. 

Other edits/comments/questions:

Line 45 discusses 5 groups of abnormalities, but only one is discussed. What are the other groups and are the other groups related to tissue hypoxia?

Line 38 "microvascular" should be replaced by "Microvasculature" 

Table 1: The description should be labeled simply as "Baseline demographics." The column with descriptors should have (years +/- SD) instead of (years), BMI should be labeled as kg/m2 +/- SD. 

In Tables 6-9, what does 5-fold cross-validation mean? This was not described in the statistical methods?

Comments on the Quality of English Language

Very minimal editing changes are needed throughout. 

Author Response

We deeply appreciate for the reviewer’s valuable comments and hints that have been very helpful to improve our revised manuscript. The revision is polished as careful and thoughtful as we can. Also, the response of your suggestions and concerns are all addressed one by one in this letter and the related descriptions are added in this revision.

Reviewer 2 Report

Comments and Suggestions for Authors

A very informative and educational manuscript that has both clinical interest and merit.  However, there are some editing issues that the authors should consider and address.  The following are suggestions/comments regarding those issues.  Line 90, "... clinical physicians better access to assess the patient's ...".  Line 106, "... to the interest in complication detection, research ...".  Line 200, "... 24 features (6 stages x 4 signals)."  Line 237, "... VV-ECMO group there were 44 data ...".  Line 238, "... the VA_ECMO group there were 41 ...".  Line 264, "... we constructed a SVM model ...".  Line 370 "... results in Figure 8 which shows a more ...".  

Author Response

(The authors gave the same response as above.)

Reviewer 3 Report

Comments and Suggestions for Authors

Despite the general design of experiment looks sound, I still do not understand what is the aim of this study and how it can improve the treatment of the patient. I suggest to clarify it overall the article - what is predicted, how it can be used for better patient treatment or reducing some risks, what the obtained result change in medical care procedure and how it can be applied into the practice.

Author Response

(The authors gave the same response as above.)

Round 2

Reviewer 3 Report

Comments and Suggestions for Authors

The article was substantially improved and can be accepted in present form.

Comments on the Quality of English Language

English language is good, except some minor typos.

Author Response

Thank you very much for the valuable feedback and for the high praise you have given us. The insights provided by the editor have been instrumental in enhancing the completeness and value of our article. Through diligent revisions and discussions with numerous experts during this period, we aim to refine the article further, making it more sophisticated and beneficial for a broader readership, especially those with a background in this field. We appreciate your patience in awaiting the completion of this manuscript. Our entire team sincerely thanks you for your support.